# Association between Outdoor and Indoor Air Pollution Sources and Atopic Eczema among Preschool Children in South Africa

**DOI:** 10.3390/ijerph21030326

**Published:** 2024-03-11

**Authors:** Mandla Bhuda, Janine Wichmann, Joyce Shirinde

**Affiliations:** 1Department of Operations Management, University of South Africa, P.O. Box 392, Pretoria 0003, South Africa; 2School of Health Systems and Public Health, Health Sciences Faculty, University of Pretoria, P.O. Box 667, Pretoria 0001, South Africa; janine.wichmann@up.ac.za (J.W.); joyce.shirinde@up.ac.za (J.S.)

**Keywords:** outdoor, indoor, air pollution, atopic eczema, South Africa

## Abstract

The objective of the study was to investigate the association between outdoor and indoor air pollution sources and atopic eczema among preschool children in South Africa. A cross-sectional design, following the International Study of Asthma and Allergies in Childhood (ISAAC) Phase III protocol, was applied. The study was conducted in Mabopane and Soshanguve Townships in the City of Tshwane Metropolitan Municipality in Gauteng, South Africa. A total population of 1844 preschool children aged 7 years and below participated in the study; 1840 were included in the final data analysis. Data were analyzed using multilevel logistic regression analysis. The prevalence of eczema ever (EE) and current eczema symptoms (ESs) was 11.9% and 13.3%, respectively. The use of open fires (paraffin, wood, or coal) for cooking and heating increased the likelihood of EE (OR = 1.63; 95% CI: 0.76–3.52) and current ESs (OR = 1.94; 95% CI: 1.00–3.74). Environmental tobacco smoke (ETS) exposure at home increased the likelihood of EE (OR = 1.66; 95% CI: 1.08–2.55) and current ESs (OR = 1.61; 95% CI: 1.07–2.43). Mothers or female guardians smoking cigarettes increased the likelihood of EE (OR = 1.50; 95% CI: 0.86–2.62) and current ESs (OR = 1.23; 95% CI: 0.71–2.13). The use of combined building materials in homes increased the likelihood of EE, and corrugated iron significantly increased the likelihood of current ESs. The frequency of trucks passing near the preschool children’s residences on weekdays was found to be associated with EE and current ESs, with a significant association observed when trucks passed the children’s residences almost all day on weekdays. Atopic eczema was positively associated with exposure to outdoor and indoor air pollution sources.

## 1. Introduction

Air pollution is a mixture of particles and gases that can reach harmful concentrations in both outdoor and indoor environments [1]. The Environment Protection Agency (EPA) recognizes six primary and secondary air pollutants: particulate matter (PM) with an aerodynamic size equal to or less than 2.5 μm and 10 μm, sulfur oxide (SOx), nitrogen oxide (NO_x_), carbon monoxide (CO), lead (Pb), and ground-level ozone (O_3_) [2]. Primary air pollutants are directly emitted from the source/activity (e.g., NOx) [1], while secondary air pollutants originate from the chemical and photochemical reactions of primary pollutants in the atmosphere (e.g., O_3_) [1]. Air pollution has various potential outdoor and indoor sources, which can be both anthropogenic and natural [1]. Anthropogenic sources include power plant generators, motor vehicles, waste incineration, and the domestic combustion of solid fuels (coal, wood, and paraffin) [3]. Natural sources of air pollution include wildfires, dust from storms, and volcanic eruptions [1]. In South Africa, common outdoor and indoor air pollutants include PM, nitrogen dioxide (NO_2_), sulfur dioxide (SO_2_), VOC, benzene, CO, and O_3_ [4,5].

The main sources of outdoor and indoor air pollution in South Africa are power plants, solid fuel combustion, transportation, environmental tobacco smoke, waste incineration, mining, and metal industries [3,6,7]. It is widely acknowledged that power plant generators play a significant role in the high levels of outdoor air pollution in South Africa [3]. This is due to the combustion of coal for electricity generation. Air pollutants associated with the combustion of coal include PM_10–2.5_, SO_2_, NOx, and mercury (Hg) [3]. Additionally, the combustion of domestic biomass fuels, such as wood and paraffin (kerosene), is also linked to high levels of indoor air pollution, such as PM, CO, and VOC [3].

The transportation sector (vehicle emissions) is linked to a high level of transport-related outdoor air pollutants such as NO_2_, CO_2_, VOC, and PM_2.5_ [8]. Waste incineration is associated with CO_2_, nitrous oxide (N_2_O), NOx, and ammonia (NH_3_) air pollutants. Exposure to ETS is associated with human carcinogenic air pollutants such as benzene [9]. Mining industry activities such as coal mining are associated with SO_2_, NO_2_, and PM_10–2.5_, CO, and O_3_ [10]. Similarly, metal industries such as steel or iron are highly associated with PM_2.5_, NOx, SO_2_, and CO [11]. Globally, outdoor and indoor air pollution continue to be a serious public health problem, accounting for approximately 7 million deaths annually [1,12]. Greenpeace’s report reveals that South Africans are exposed to air pollution levels exceeding World Health Organization (WHO) air quality standards [13]. Although there are air quality regulations to manage the criteria for air pollutants in South Africa, air pollution continues to be a public health concern. Recent studies conducted in South Africa have reported air pollution levels (e.g., PM_2.5_) that have exceeded the WHO air quality guidelines and the South African National Quality Standards [14,15].

South Africa is ranked 4th on the African continent for the highest number of deaths associated with exposure to air pollution [16]. Exposure to air pollutants has been linked to respiratory illness, cardiovascular diseases, and allergies in epidemiology studies conducted locally [11,17,18,19,20,21,22]. In a case-crossover study conducted by Thabethe et al. [23], a positive association was found between air pollution exposure and deaths in three major South African cities (Cape Town, Johannesburg, and Durban). Children are more vulnerable to the health effects of air pollution due to their immature immune systems and underdeveloped lungs [24]. In addition, children spend most of their time indoors with their mothers doing domestic chores, such as preparing food and cleaning [25]. Therefore, susceptible groups such as children should be prioritized to mitigate the adverse health effects associated with exposure to outdoor and indoor air pollution sources, in addition to achieving Sustainable Development Goal (SDG) 3 on good health and well-being [26].

Atopic eczema (AE) (synonymous with atopic dermatitis and eczema) is described as a chronic inflammatory skin disease that frequently precedes the development of asthma or other respiratory allergies that primarily occur in early childhood [27,28,29]. According to the global report on atopic dermatitis, an estimated 223 million people suffer from AE [30]. Of these, 20% and 10% are children and adults, respectively [30]. The prevalence of AE is more commonly found in urban areas than in rural areas [31]. Studies have reported a growing occurrence of AE in developing countries [32,33]. The reason for the increase in the prevalence of AE is not clearly understood, including its pathophysiology [1]. The disease is characterized by skin inflammation of T-helper 2 cells, chronic pruritus, and disruption of the skin barrier due to the increased penetration of environmental contaminants through the stratum corneum [1,34]. Patients with AE tend to have increased levels of total immunoglobulin E (IgE) [1]. Although both genetic and environmental factors contribute to the development of AE, recent studies suggest that the rise in the prevalence of AE is mainly due to environmental factors [1,30]. Humans are exposed to environmental contaminants, such as indoor and outdoor air pollution, through three routes of entry: inhalation, ingestion, and dermal contact [35]. However, exposure to air pollution through skin contact has been linked to immune reactions that can result in allergic responses, which in turn can exacerbate the symptoms of AE [36]. A recent review by Fadadu et al. [37] suggests that air pollutants cause oxidative stress, which triggers a proinflammatory response, activates the aryl hydrocarbon receptor system, and compromises the integrity of the skin barrier, all of which exacerbates AE symptoms and disease.

Common symptoms associated with AE include dry skin, localized red scaly patches, intense itching, and skin pain [30]. The symptoms, while not life-threatening, can negatively impact individuals’ quality of life, including their ability to sleep [30,31,38]. The management of AE symptoms includes the use of oral medication, moisturizers, and systematic immune suppressors, depending on the severity [39]. The clinical diagnosis of AE differs depending on the age group (infants, children, or adults) [31]. However, the clinical diagnosis process in Africa is still evolving due to limited access to well-resourced healthcare centers [31]. This leads to under-reporting of the disease. Recent epidemiology studies have linked outdoor and indoor air pollution sources such as ETS, open fires (wood/paraffin/coal), and vehicle emissions with AE symptoms in younger children and adolescents in developed and developing countries [27,29,40,41,42,43,44,45,46,47,48,49,50]. For instance, a cross-sectional study conducted in 10 cities in China found that cooking with biomass fuel (coal or wood) increased the risk of eczema by 40% in children aged 3 to 6 years old [43]. In addition, exposure to indoor dust mites, animal dander, and mold spores has been associated with AE symptoms [29,50].

A time series study conducted in China involving 214,747 children found a positive association between AE and exposure to PM_2.5–10_, SO_2_, and NO_2_ during the study period [51]. A panel study conducted in metropolitan Seoul, Korea, among 117 children under the age of 5 years old with AE reported a positive association with exposure to PM_10_, NO_2_, and O_3_ levels [52]. A systematic analysis including 12 epidemiology studies reported a positive association between maternal exposure to NO_2_ and childhood eczema [20]. A positive association was found between symptoms of AE and exposure to butyl benzyl phthalate (VOC dust) reported by a nested case-control study of 10,852 children aged 3 to 8 years [53]. There are fewer epidemiology studies conducted in low- and middle-income countries, such as South Africa, focusing on the association between air pollution and AE in preschool children, which may be due to the belief that AE cannot be influenced by air pollution. Therefore, there are few available data regarding the incidence of AE in association with air pollution in South Africa [31]. To the best of our knowledge, this is the first analytical cross-sectional study conducted in selected townships located in the City of Tshwane Metropolitan Municipality, Gauteng Province, South Africa, focusing on preschool children aged 7 years old and below. The study aimed to investigate the association between outdoor and indoor air pollution sources and atopic eczema among preschool children in South Africa.

## 2. Methods

### 2.1. Study Setting

The study was conducted in Mabopane and Soshanguve Townships in the City of Tshwane Metropolitan Municipality in Gauteng, South Africa. Mabopane is located at 25°29′51″ S, 28°06′02″ E, while Soshanguve is located at −2530′53″ S, 2806′29″ E. According to the South African Census 2011, the population of Mabopane and Soshanguve was estimated at 110,972 and 403,162, respectively, with most people in both townships being black [54]. Setswana is the most spoken language in Mabopane (58.8%), while Sepedi is the most spoken language in Soshanguve [54]. According to Statistics South Africa, 15.9% and 16.5% of the population in Mabopane and Soshanguve, respectively, have no source of income [54].

### 2.2. Study Design, Population, and Sample Selection

The study applied an analytical cross-sectional design, following the ISAAC Phase III protocol. The study was conducted from January 2022 to March 2023, focusing on children aged 7 years and younger attending preschools or daycare centers in Mabopane and Soshanguve townships. A total of 42 preschools were randomly selected from the preschool list provided by the Gauteng Department of Social Development. Each preschool was contacted and requested to participate in the study, and upon approval from the preschool principal, all eligible children’s parents or guardians participated. The English version of the ISAAC written questionnaire was used to collect data, which was distributed to parents or guardians by the selected preschool principals or owners for completion on behalf of the children. A total of 1844 preschool children’s parents and guardians completed the ISAAC questionnaire and returned it to the preschool for collection. However, 1840 participants met the study inclusion criteria (7-year-old and younger children residing and attending preschools in the study setting whose parents or guardians had signed consent forms) and were included in the final data analysis.

### 2.3. Questionnaire

The ISAAC Phase III protocol study questionnaire with the standardized core questionnaire on demographics and asthma symptoms was used to collect the data. Information on potential risk factors was obtained with questions about current living conditions, socioeconomic status, environmental tobacco smoke, cooking and heating fuels, pet ownership, and the number of years lived in the community. The parents/guardians were asked to indicate their response by ticking or writing in the appropriate box provided for each question. Other similar studies have used the ISAAC questionnaire [44,55,56].

### 2.4. Health Outcomes, Air Pollution Sources, and Confounders

#### 2.4.1. Health Outcomes

Current ES was estimated based on positive answers given by parents or guardians to the following three written questionnaire questions: (1)Has your child ever had an itchy rash that was coming and going for the past 6 months? (Yes\No)(2)Has your child had this itchy rash at any time in the past 12 months? (Yes\No)(3)Has this itchy rash at any time affected your child on any of the following places: the folds of the elbow, behind the knees, in front of the ankles, under the buttocks, or around the neck, ears, or eyes? (Yes\No)
EE was estimated based on positive answers given by parents or guardians to the following written questionnaire question:Has your child ever had eczema? (Yes\No).

#### 2.4.2. Air Pollution Sources

Air pollution sources included ETS exposure at home in the past 30 days (yes/no), ETS exposure at school in the past 30 days (yes/no), a mother or father smoking tobacco (yes/no), or any other person smoking tobacco at home other than parents (yes/no). The parents or guardian children were asked to select the most frequently used source of energy at home, and therefore they had to select only one type of energy source for cooking and heating (electricity/gas/paraffin/wood/coal). They were also asked to state the frequency of trucks passing near residences (never/seldom/frequently through the day/almost all day).

#### 2.4.3. Confounders

Potential confounders in the study included sex (female or male), hours spent watching television per day, type of house (brick, mud, corrugated iron, combination), mode of transport to preschool (walking, taxi/bus, car, combination of car/taxi or train), and child being born in the townships (Soshanguve and Mabopane).

### 2.5. Data Management and Statistical Analysis

The data were captured in a database using EPI Data. Stata version 15 (STATA Corp, Texas, USA) was used to analyze the data. The prevalence of the health outcomes, the proportion of air pollution sources under investigation, and confounding variables were calculated by dividing the number of participants who responded affirmatively to a particular question by the number of questionnaires completed. Observations marked as “do not know”, “not stated”, or “other responses” were set as missing but were included in the descriptive analyses. Therefore, each question had a slightly different sample size.

A chi-squared test was applied to determine the relationship between Mabopane/Soshanguve and the confounding variables. Crude and adjusted odds ratios (ORs) and 95% confidence intervals (Cis) were calculated to estimate the likelihood of health outcomes given the presence of an air pollution source and confounding variables. Univariate and multiple logistic regression analysis (MLRA) were applied. Not all children were ≤7 years old at the time of the survey. Therefore, the data analysis was restricted to ≤7 years old children residing in Mabopane and Soshanguve. As a result, each MLR model had a different sample size. Air pollution sources and confounding variables with *p* < 0.20 obtained in the univariate logistic regression analysis (LRA) were included in the MLRA. A *p*-value of less than 0.05 in the MLRA was considered statistically significant.

## 3. Results

### 3.1. Profile of the Study Participants

A total of 1844 parents or child guardians completed the self-administered ISAAC questionnaire on behalf of their children. Table 1 summarizes the demographic characteristics and household conditions, along with the prevalence of current eczema symptoms (ESs) and the prevalence of ever having had eczema (EE). In total, 56% of the participants were from Soshanguve. The majority of the children were female (53.6%). Overall, 46% of the participants preferred spending time indoors watching TV for more than 3 h. The majority of the participants had access to electricity for cooking and heating purposes (88.4%). In total, 44.5% of the study population confirmed that trucks seldom passed near their residence on weekdays. The majority (76.8%) of the participants had not been exposed to ETS at home in the past 30 days. In addition, 82.3% of the participants had not been exposed to ETS at school in the past 30 days, while 88% of the participants were born in the study areas. The majority of the participants lived in formal housing structures (86.5%). A total of 45.3% of the children walked to their preschool.

### 3.2. Prevalance of Eczema Ever (EE) and Current Eczema Symptoms (ESs)

Appendix A summarizes the prevalence of eczema ever (EE) and current eczema symptoms (ESs) by sex (male and female). The prevalence of EE in female and males was 10.55% (n = 986) and 11.16% (n = 851), respectively. The prevalence of current ESs in females and males was 13.18% (n = 986) and 13.63% (n = 851), respectively. Appendix A summarizes the overall prevalence of EE and current ESs. The overall study prevalence of EE and current ESs was 11.9% (n = 119) and 13.3% (n = 246), respectively, with a total of 1840 participants.

### 3.3. Univariate and Multiple Logistic Regression Analysis (MLRA) of Eczema Ever and Potential Risk Factors

Table 2 summarizes the results of univariate analysis and MLRA for EE and potential risk factors. Males were more likely to have EE (OR = 1.01; 95% CI: 0.75–1.38). However, spending hours watching television was a protective factor. Using open fires (paraffin/wood/coal) for cooking and heating increased the likelihood of EE (OR = 1.63; 95% CI: 0.76–3.52). ETS exposure at home increased the likelihood of EE (OR = 1.66; 95% CI: 1.08–2.55). Cigarette smoking by the mother or female guardian (OR = 1.50; 95% CI: 0.86–2.62) and the father or male guardian (OR = 1.07; 95% CI: 0.73–1.57) increased the likelihood of EE. The type of housing increased the likelihood of EE, as seen in the figures for combination (OR = 2.34; 95% CI: 1.33–4.13) and mud (OR = 1.13; 95% CI: 0.14–9.54) construction; however, the association was significantly detrimental in the combined house type. The mode of transport to preschool was detrimentally associated with EE as follows: motor car (OR = 3.64; 95% CI: 1.74–7.51), taxi/bus (OR = 3.90; 95% Cl: 1.84–8.25), walking (OR = 2.73; 95% Cl: 1.32–5.61), and other (OR = 4.86; 95% CI: 15.32). The frequency of trucks passing near residences on weekdays almost all day (OR = 1.98; 95% CI: 1.08–3.61) and seldomly (OR = 1.61; 95% CI: 1.08–2.39) significantly increased the likelihood of EE.

### 3.4. Univariate and Multiple Logistic Regression Analysis (MLRA) of Current Eczema Symptoms and Potential Risk Factors

Appendix A summarizes the results of the MLRA for current ESs and potential risk factors. Sex (male or female) and spending hours watching television were protective factors for current ESs. Using gas (OR = 1.63; 95% CI: 1.00–2.65) and open fires (paraffin/wood/coal) (OR = 1.94–95% CI: 1.00–3.74) increased the likelihood of current ESs. ETS exposure at home increased the likelihood of current ESs (OR = 1.61; 95% CI: 1.07–2.43). Cigarette smoking by the mother or female guardian increased the likelihood of current ESs (OR = 1.50; 95% CI: 0.86–2.62). However, there was no association between cigarette smoking by the father or male guardian and current ESs. A corrugated iron house type (OR = 1.64; 95% Cl = 0.99–2.73) increased the likelihood of current ESs. The likelihood of current ESs was significantly increased by the mode of transport used to get to preschool as follows: motor car (OR = 2.27; 95% CI: 1.19–4.34), taxi/bus (OR = 3.11; 95% CI: 1.63; 5.95), walk (OR = 2.68; 95% CI: 1.46; 4.95), and others (OR = 4.57; 95% CI: 1.67–12.51). The frequency of trucks passing near residences on weekdays almost all day (OR = 2.08; 95% CI: 1.24–3.49) and seldomly (OR = 1.15; 95% Cl: 0.81–1.64) increased the likelihood of current ESs.

## 4. Discussion

The study investigated the association between outdoor and indoor air pollution and atopic eczema among preschool children in South Africa. The prevalence of EE and current ESs was slightly different in females and males, with the males reporting a higher prevalence of EE and current Ess, although more study participants were females compared to males. The two townships have a similar sociodemographic status and are located less than 10 km apart from each other. In this study, the overall prevalence of EE (11.9%) and current ESs (13.37%) was found to be lower than in other studies conducted in Africa, such as Angola, where EE was reported at 18.4% among 6- to 7-year-old children (Ref. [57]) and Ethiopia, where AE was reported at 21% among children under 12 years old [58]. It is worth noting that South Africa has a larger economy than Angola and Ethiopia [59], with the highest consumption of coal on the African continent in 2023.

A cross-sectional study following the Phase III ISAAC protocol, conducted in Limpopo Province, South Africa among 2437 children aged from 6 to 7 years old, reported a higher prevalence of EE and ESs of 38% and 17.1%, respectively [60]. The distance between Limpopo Province and the study area (Gauteng Province) is about 400 km, and Limpopo Province is more rural and surrounded by active mining activities and agricultural activities. Another cross-sectional study following the Phase I ISAAC protocol conducted in the air pollution priority area (City of Ekurhuleni Metropolitan Municipality) in Gauteng Province, South Africa, reported a higher prevalence of EE (14%) among 3424 adolescents aged 13 and 14 years old; however, the study reported a lower prevalence of current ESs (9.6%) compared to our findings [40]. Subsequently, a recent Global Asthma Network (GAN) study, including 3957 adolescents aged 13 to 14 years old in KwaZulu Natal Province, South Africa, reported a lower prevalence of AE (7.1%) compared to our study [61].

Another study conducted in Westen Cape Province, South Africa, including children aged less than 12 months (<1 year old) and 1 to 5 years old reported a higher AE prevalence of 43% and 67%, respectively, compared to our results. [39]. A retrospective review by Kakande et al. [62] reported a higher prevalence of AE among children who attended the dermatology outpatient clinic in the Western Cape Province, South Africa, between December 2005 and December 2010. The Western Cape Province is more affluent compared to KwaZulu Natal, Gauteng, and Limpopo Provinces in South Africa. It is evident that the prevalence of AE and its symptoms remains a challenge, independent of the socioeconomic dynamics of the geographic environment.

There was no significant association between EE and current ESs and the participants’ sex (*p* > 0.005). Our findings were consistent with the ISAAC cross-sectional study, which included children aged 6 to 7 years in Angola and reported no association between sex and AE symptoms [57]. It is uncertain whether gender or sex makes young children more susceptible to eczema symptoms. In contrast to our findings, two studies conducted in China among 5730 children aged 3 to 8 years old and in Saudi Arabia among 964 preschool children aged 2 to 8 years old found sex was a risk factor for eczema [43,50]. It is unclear whether sex or gender predisposes preschool children to AE symptoms. Watching television for 3 h or more was significantly associated with current ESs among the preschool children in the study setting (*p* < 0.005). Another ISAAC Phase III cross-sectional study, which included 29 centers across 17 countries, reported that adolescents aged 13 and 14 years old who watched television for more than three hours had a positive association with current ESs [63]. Therefore, spending more time indoors and watching television increases the likelihood of current ESs in children and adolescents due to their exposure to potential indoor air pollution sources.

The use of gas for domestic purpose was associated with current ESs in the study; the results are consistent with the cross-sectional study done in Gauteng, South Africa, including 12- to 14-year-old adolescents, where gas was significantly associated with EE [40]. A meta-analysis comprising 12 epidemiological studies revealed a relationship between prenatal NO_2_ exposure and eczema risk: for every 10 μg·m^−3^ increase in maternal exposure to NO_2_ throughout the course of the pregnancy, there would be a 13% increase in the risk of childhood eczema [20]. According to a time series study conducted in Chongqing, China, among 214,747 children aged 0 to 18 years old, there was a significant correlation between AE symptoms and every 1 mg·m^−3^ increase in CO level [51].

The use of open fire sources (wood, coal, and kerosene) for domestic purposes (heating and cooking) was found to be associated with the prevalence of EE and current ESs. Our findings are consistent with a Phase III ISAAC cross-sectional study including 47 countries, where the use of open fires for cooking was associated with a reported association with current ESs among 6 to 7-year-old children; in addition, EE and current ESs were associated with the use of open fires among 13 to 14-year-old [64]. Subsequently, the use of open fire sources (wood and coal) was reported to increase the likelihood of AE by 40% among 5730 children aged 3–6-years old and 7–8 years old in ten cities in China [43].

According to a panel study involving 177 AE patients in Korea, children under the age of 5 years old were more likely to experience AE symptoms when levels of PM_10_, NO_2_, and O_3_ increased by 10 units [52]. Despite the fact that about 93% and 91% of the residents of Mabopane and Soshanguve have access to electricity, respectively [65]; they still use open fire sources (coal, wood, and paraffin) for domestic purposes. This may be due to the rising costs of electricity and continued load-shedding in low-income countries such as South Africa [66,67,68].

Environmental tobacco smoke in children’s homes was positively associated with EE and current ESs. In addition, cigarette smoking by the mother or female guardian was associated with EE among the preschool children; however, the association was not significant (*p* > 0.005). Our results are consistent with other cross-sectional studies conducted in the rural area of Limpopo and the urban area of Gauteng, South Africa, including younger children (6 to 7 years old) and adolescents (13 to 14 years old) [40,44]. Another Phase III ISAAC study, including 75 centers in 32 countries, reported a positive association between current ESs among 220,407 children aged 6 to 7 years old and 350,654 adolescents aged 13 to 14 years old with cigarette smoking by maternal and paternal [69]. The findings are consistent with a larger cross-sectional study including 145,702 participants aged from 12 to 18 years old who reported a positive association between AE and tobacco smoke (active and passive smoking) [70]. Another cross-sectional study from six French communities reported a positive association between exposure to pollutants associated with ETS, such as benzene and AE, among 4907 children aged 9 to 11 years old [71].

Exposure to tobacco smoke continues to be a public health problem, which accounts for about 1.3 million deaths annually due to exposure to second-hand tobacco smoke worldwide [72]. According to the South Africa Demographic and Health Survey, the prevalence of tobacco in South Africa was 7% and 37% for women and men, respectively [73]. Therefore, the current South African tobacco regulations should be enforced to prevent tobacco smoking in the indoor environment. The government should promote a health education campaign to educate the public, including children’s parents, about the health effects associated with indoor and outdoor tobacco smoke, such as AE. However, a new tobacco products and electronic delivery systems bill of 2022 was introduced in South Africa to institute total non-smoking in indoor and some designated outdoor places.

The prevalence of EE and present ESs among study participants was positively associated with the type of household. According to our findings, children from households built with a combination of building materials—corrugated iron, mud, and brick—were more likely to have both EE and current ESs. Therefore, a child’s development of AE symptoms is greatly influenced by the type of environment in which they live.

Children’s modes of transportation, including walking and the use of private motor vehicles or public vehicles (taxis or buses), play a crucial role in the development of EE and current ESs among preschool children aged 7 years old and below. For instance, we observed a significant association between the use of public motor vehicles (taxis or buses) and EE and current ESs among the preschool children in our study. In addition, the frequency of trucks passing near the preschool children’s residences on weekdays was found to be associated with EE and current ESs, with a significant association observed when trucks pass the children’s residences almost all day during the weekdays. Our results are consistent with the ISAAC study conducted in the same province (Gauteng), including 12- to 14-year-olds [40]. Contrary to the study conducted in Skopje, Republic of Macedonia, including adolescents aged 13 to 14 years old, no association was found between the frequency of trucks passing near the participants’ homes and AE [48].

A study conducted among 293,343 patients with AE in Guangdong Province, Southeastern China, using a time series study design, reported a positive association between exposure to a combination of traffic-related ambient air pollutants, including NO_2_, PM_2.5–10_, and SO_3_, [74]. In South Africa, it was estimated that transportation is responsible for 90% of transport-related emissions (e.g., CO, NO_2_, etc.) due to an increased number of privately owned vehicles [3,75], although there are strategies in South Africa, such as the green transport strategy, aimed at mitigating transport-related air pollutants such as CO. Therefore, the country should invest in the implementation of strategies to cut emissions of transport air pollution. In addition, mixed modes of transportation should be encouraged to mitigate transport-related air pollutants.

Some limitations should be considered when interpreting the study. Firstly, a cross-sectional study design was used. Cross-sectional studies are weak at proving causation as they are subject to difficulties in the interpretation of the temporal sequence of events. Secondly, self-reporting questionnaires were used, which may lead to the misclassification of health outcomes and exposure status, resulting in statistical significance. Thirdly, no indoor and outdoor exposure assessments were carried out to quantify the levels of the pollutants. Fourthly, no clinical tests were carried out on the participants. Lastly, the questionnaires were not translated into other dominant languages in the study setting. The strength of our study includes the use of a validated ISAAC questionnaire, which has been used in many studies globally with consistent results.

## 5. Conclusions

Although the prevalence of EE and current ESs in the study setting was relatively low compared to the total number of participants, the study has provided preliminary results on the association between outdoor and indoor air pollution sources such as open fires (coal, wood, or paraffin), ETS at home, modes of transport to preschools, and proximity to busy roads with atopic eczema among preschool children in South Africa, a low–middle-income country. The findings should serve as scientific evidence for South African policymakers to develop and strengthen current air pollution levels to prevent health outcomes associated with outdoor and indoor air pollution sources, such as atopic eczema. In addition, the study serves as a foundation for further research on the link between specific air pollutants and atopic eczema in preschool children.

## Figures and Tables

**Table 1 ijerph-21-00326-t001:** Demographic characteristics and household conditions (n = 1840).

	Total	Percentage
Place of residence		
Mabopane	801	43.6
Soshanguve	1039	56.4
Sex		
Female	986	53.6
Male	851	46.2
Missing	3	0.2
Hours spent watching television per week		
Less than 1 h	322	17.5
1 h to less than 3 h	620	33.7
3 h or more	857	46.6
Missing	41	2.2
Residential cooking/heating fuel type		
Electricity	1628	88.5
Gas	115	6.2
Open fire (paraffin/wood/coal)	61	3.3
Missing	36	2.0
Frequency of trucks passing near residence on weekdays		
Almost all day	124	6.7
Frequently through the day	406	22.1
Never	454	24.7
Seldom	819	44.5
Missing	37	2.0
ETS exposure at home in the past 30 days		
No	1414	76.8
Yes	306	16.6
Missing	120	6.5
ETS exposure at school in the past 30 days		
No	1515	82.3
Yes	25	1.4
Missing	300	16.3
Born in the areas of Mabopane and Soshanguve		
No	212	11.9
Yes	1574	88.1
Type of house		
Brick	1570	86.5
Combination	87	4.8
Corrugated iron	114	6.3
Mud	10	0.6
Other	33	1.8
Mode of transport to school		
Combination	230	12.7
Motor car	384	21.1
Taxi/bus	346	19.0
Walk	823	45.3
Other	34	1.9

**Table 2 ijerph-21-00326-t002:** The prevalence of ever having had eczema (EE) among the participants, along with crude and adjusted ORs.

Variable	Total * (%)	Crude OR (95% CI) ^†^	*p*-Value	Adjusted OR (95% CI) ^†^	*p*-Value
**Sex**					
Female	104 (52.3)	1		1	
Male	95 (47.7)	1.07 (0.79–1.43)	0.672	1.01 (0.75–1.38)	0.925
**Hours spent watching television per week**					
Less than 1 h	42 (21.1)	1		1	
1 h to less than 3 h	77 (38.7)	0.95 (0.63–1.41)	0.784	0.92 (0.60–1.40)	0.694
3 h or more	78 (39.2)	0.67 (0.45; 1.00)	0.047	0.58 (0.38–0.90)	**0.014**
**Residential cooking/heating fuel type**					
Electricity	172 (86.4)	1		1	
Gas	14 (7.0)	1.17 (0.66–2.10)	0.590	0.92 (0.5; 1.69)	0.789
Open fire (paraffin/wood/coal)	10 (5.0)	1.66 (0.83–3.33)	0.154	1.63 (0.76–3.52)	0.211
**Smoking exposure at home in the past 30 days**					
No	136 (68.3)	1		1	
Yes	54 (27.1)	2.01 (1.43–2.84)	**<0.001**	1.66 (1.08; 2.55)	**0.021**
**Smoking exposure at school in the past 30 days**					
No	157 (78.9)	1		1	
Yes	2 (1.0)	0.75 (0.1–3.22)	0.701	0.39 (0.09–1.79)	0.227
**Mother/female guardian smoke cigarettes**					
No	180 (90.5)	1		1	
Yes	19 (9.5)	1.75 (1.04–2.94)	0.034	1.50 (0.86–2.62)	0.157
**Father/male guardian smoke cigarettes**					
No	136 (68.3)	1		1	
Yes	63 (31.7)	1.37 (1.00–1.89)	0.051	1.07 (0.73–1.57)	0.736
**Type of house**					
Brick	164 (82.4)	1		1	
Combination	19 (9.5)	2.40 (1.40–4.08)	**0.001**	2.34 (1.33–4.13)	**0.003**
Corrugated iron	11 (5.5)	0.92 (0.48–1.74)	0.788	0.73 (0.37–1.45)	0.371
Mud	1 (0.5)	0.95 (0.12–7.57)	0.963	1.13 (0.14–9.54)	0.907
Other	4 (2.0)	1.18 (0.41–3.41)	0.756	0.98 (0.31–3.10)	0.977
**Mode of transport to school**					
Combination	9 (4.5)	1		1	
Motor car	54 (27.1)	4.02 (1.94–8.30)	**<0.001**	3.64 (1.74; 7.61)	**<0.001**
Taxi/bus	48 (24.1)	3.96 (1.90–8.23)	**<0.001**	3.90 (1.84; 8.25)	**<0.001**
Walk	82 (41.2)	2.72 (1.34–5.50)	**0.005**	2.73 (1.32; 5.61)	**0.006**
Other	6 (3.0)	5.26 (1.74–15.89)	**0.003**	4.86 (1.54–15.32)	**0.007**
**Frequency of trucks passing near residence on weekdays**					
Never	40 (20.1)	1		1	
Almost all day	20 (10.1)	1.99 (1.12–3.55)	**0.020**	1.98 (1.08–3.61)	**0.027**
Frequently through the day	27 (13.6)	0.74 (0.44–1.23)	0.239	0.73 (0.44–1.24)	0.245
Seldom	110 (55.3)	1.61 (1.10–2.35)	**0.015**	1.61 (1.08–2.39)	**0.019**

* Total for each risk factor is different due to differences in missing values. ^†^ Model adjusted for all the variables. Values that are statistically significant at less than 0.02 for the crude OR and less than 0.05 for the adjusted OR are in bold font. EE: Eczema Ever.

## Data Availability

We did not receive ethics approval to share the raw field data publicly. The data belong to the University of Pretoria (UP). The raw data analyzed in the current study are available from UP on reasonable request.

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
