# Peer review of "Association between Outdoor and Indoor Air Pollution Sources and Atopic Eczema among Preschool Children in South Africa"

_ijerph, 2024, doi:10.3390/ijerph21030326_

Round 1
Reviewer 1 Report
Comments and Suggestions for Authors
Bhuda and colleagues aim to investigate the associations between sources of outdoor and indoor air pollution and the incidences of atopic eczema in children aged 7 years old and below in South Africa. The study enrolled 1840 volunteers in the cross-sectional study. The study indicated that atopic eczema was positively associated with exposure to out- 316 door and indoor air pollution sources. The authors also elaborated on the limitations of this study in the manuscript. The manuscript seems well-organized. Yet, the manuscript concealed two major issues.
- The study should demonstrate the causal relationship between air pollutants and the incidences of atopic eczema in children aged 7 years old and below. Which air pollutants have been documented resulting in the incidences of atopic eczema in children aged 7 years old and below? In lines 47-50, the authors claimed the associations between exposure to indoor and outdoor air pollution sources and atopic eczema in younger children and adolescents with references 12-23. Some references are reviews, while others investigated the impacts of environmental factors on incidences of atopic eczema in younger children and adolescents. The author should demonstrate the causal relationship between specific air pollutants and the incidences of atopic eczema clearly.
- If a causal relationship exists between air pollutants and the incidences of atopic eczema in children aged 7 years old and below, the authors should discuss the potential mechanism that physiological and pathological reasons leading to the incidences of atopic eczema induced by specific air pollutants in the introduction and discussion section. If these two major issues are not addressed, there is a lack of theoretical basis in supporting the hypothesis of this study.
Minor issues
Line 236: Remove the underline for the references.
Line 282: Remove the underline for the references.
Figure: move Table 2 or Table 3 to the supplementary material. Use figure instead.
Author Response
Good day,
Thank you very much for the review!
Please see the attached responses.
Kind regads,
Bhuda

Reviewer 2 Report
Comments and Suggestions for Authors
- place abbreviations at the first moment the term is mentioned in the text (highlighted throughout the text), preferably in the introduction, if there is a word limit in the abstract;
- line 148: I did not find the abbreviation for LRA, only for MLRA;
- maybe add a pie chart for the eczema data, making it more attractive to the reader;
- line 235: The study con-234 conducted in China and South Arabia reported an association between eczema in children 235 and male children [16, 44]. It was not clear whether the phrase refers to both sexes or just the male sex, if it refers to the male sex, is there a greater predisposition? This was not observed in the study, but does it exist?
- line 236: The use of gas was 237 associated with current ES in the study. Improve the discussion as a whole, giving more emphasis to the results observed in the study in question. This way, there was just a comparison.

Author Response
Good day,
Thank you very much for the review.
Please see the attached responses.
Kind regards,
Mandla

Reviewer 3 Report
Comments and Suggestions for Authors
In this manuscript, using a cross-sectional design, the authors investigated the associations between outdoor (such as transportation) and indoor (such as environmental tobacco smoking and cooking) air pollution sources and atopic eczema (AE) among preschool children in South Africa. They found that AE was positively associate with exposure to outdoor and indoor air pollution sources. Besides the limitations mentioned in the manuscript, from my opinion, the authors must address the following questions before being accepted for publication:
1. There are so many editorial errors in the manuscript. Please check the spelling, grammar, typo, font size, and other edits more carefully, e.g., most of the abbreviations such as EE, ES, ETS, AE, ISAAC, etc. should use their full names when they first appear (same rule for the abstract part); “tuck” in line 159 should be “truck”; lines 236 and 248, the font size are not same; line 282 underline marker; …….
2. In lines 24-26, the authors mentioned that “Frequently of trucks passing near the residence throughout the day and almost all day increased the likelihood of EE and current ES, respectively.” This looks contradict with the conclusion in lines 300-301 “However, there was no association between the frequency of trucks passing near resi-300 dences on weekdays through the day with EE and current ES”. Please clarify it.
3. The discussion part is very hard to follow. It needs significant improvement. I would suggest the authors focus on their own data/findings and do more analysis rather than simply list the results from references.
4. The authors only investigated the inhalation pathway due to exposure to possible air pollution resources. It would be very valuable to add discussions through other pathways like ingestion and dermal contact.
5. Please explain how the authors distinguish the AE, EE, and ES from the responses of the questionnaire?
Comments on the Quality of English Language
Language is fine. Minor edits are needed.
Author Response
Good day,
Thank you very much for the review!
Please see the attached responses.
Kind regards,
Mandla

Reviewer 4 Report
Comments and Suggestions for Authors
Thank you for giving me the opportunity to review this wonderful research. I find the research relevant and compelling. The extent of data collection is robust, covering 1844 preschool children aged 7 years and below. Overall, the manuscript followed proper methodology. Still, I have some minor observations, that will make this article more suitable for the international readership of IJERPF.
1. introduction should have a section on pollution sources, and waste management. Authors can start the introduction part with focus on how pollution air) is impacting lives and livelihoods across the globe, citing some recent publications. A rapid search of literature revealed articles such as, https://www.nejm.org/doi/full/10.1056/NEJMsa2300523; https://www.sciencedirect.com/science/article/abs/pii/S2214785323034119; https://www.bmj.com/content/383/bmj-2023-077784 etc, that talks about global mortality due to air pollution and its management. Then the authors should introduce their case study and study area focus to South Africa, otherwise, the article seems to have more of a regional focus.
2. Line number 62 "The prevalence of AE is more common in urban than rural settings" need a citation.
3. The questionier format can be uploaded as supplementary material to add more replicability to this study.
4. As per table 1, 88.1% of the respondents reported not having eczema in life. Then how, authors can deduce the results with only 199 individuals reporting eczema cases? Isn't this making the effective sample size smaller?
5. More discussion should be on the linkage between air pollution and eczema.
I will invite the authors to respond to these comments before recommending publication. Overall I find the research novel and worthy of consideration at IJERPH.
Comments on the Quality of English Language
Need moderate revision
Author Response

(The authors gave the same response as above.)

Round 2
Reviewer 1 Report
Comments and Suggestions for Authors
accept